# Dietary Supplemental Chromium Yeast Improved the Antioxidant Capacity, Immunity and Liver Health in Broilers under High Stocking Density

**DOI:** 10.3390/ani12172216

**Published:** 2022-08-28

**Authors:** Xiangqi Xin, Miaomiao Han, Yuan Wu, Yuanyang Dong, Zhiqiang Miao, Junzhen Zhang, Xianyi Song, Ru Jia, Yuan Su, Ci Liu, Rui Bai, Jianhui Li

**Affiliations:** College of Animal Sciences, Shanxi Agricultural University, Taigu, Jinzhong 030801, China

**Keywords:** chromium yeast, stocking density, antioxidant capacity, immunity, liver health, broilers

## Abstract

**Simple Summary:**

High stocking density can reduce the health status of broilers. Organic chromium can promote growth, immunity, antioxidant activity and alleviate stress. This study investigated the effects of different levels of yeast chromium on growth performance, organ index, antioxidant capacity, immune performance and histopathology of broilers under high stocking density and suggested that the optimal yeast chromium supplementation for antioxidant capacity was 425.00–665.00 µg Cr/kg; the optimal yeast chromium supplementation for immunity was 319.30–961.00 µg Cr/kg; and the optimal yeast chromium supplementation of liver health was 800.00–1531.60 µg Cr/kg.

**Abstract:**

This study was conducted to investigate the effects of different levels of yeast chromium on growth performance, organ index, antioxidant capacity, immune performance and liver health of broilers under high stocking density. A total of 684 1-day-old Arbor Acres broilers were selected and fed a common diet from 1 to 22 days of age. At the end of 22 days, broilers with similar weight were randomly divided into six treatments, with six replications in each treatment. The broilers in control groups were fed with a control diet and raised at low stocking density of broilers (14 broilers/m^2^, LSD) and high stocking density (20 broilers/m^2^, HSD). The broilers in treatment groups were fed with diets supplemented with 200, 400, 800 and 1600 µg Cr/kg chromium yeast (Cr-yeast) under HSD, respectively. The experimental period was 23~42 days. Compared with the LSD group, the HSD group significantly decreased the liver index (ratio of liver weight to live weight of broilers) of broilers (*p* < 0.05), the HSD group significantly increased the content of corticosterone (CORT) and the activities of alanine aminotransferase (ALT) and alkaline phosphatase (ALP) and decreased the prealbumin (PA) level in the serum (*p* < 0.05). HSD decreased the total antioxidant capacity (T-AOC) contents in the serum, liver and breast, serum glutathione peroxidase (GSH-Px) activities, breast total superoxide dismutase (T-SOD) activities and liver catalase (CAT) activities of broilers (*p* < 0.05). The HSD group significantly increased the total histopathological score (*p* < 0.05). Compared with the HSD group, adding 200, 400, and 1600 Cr-yeast significantly increased the liver index of broilers (*p* < 0.05), all HSD + Cr-yeast groups decreased the ALT activities (*p* < 0.05), and the HSD + 800 group significantly decreased the CORT contents and the ALP activities of the serum (*p* < 0.05); the HSD + 400, 800 and 1600 groups increased the PA contents of the serum (*p* < 0.05); HSD + 800 group significantly reduced the tumor necrosis factor-α (TNF-α) and Interleukin-1β (IL-1β) contents of the serum (*p* < 0.05); moreover, the HSD + 400 group increased the GSH-Px activities of the serum (*p* < 0.05), the T-AOC and the T-SOD activities of the breast (*p* < 0.05) and the T-AOC and CAT activities of the liver (*p* < 0.05). Adding 800 Cr-yeast significantly decreased the total histopathological score (degree of hepatocyte edema and inflammatory cell infiltration) under HSD (*p* < 0.05). In summary, Cr-yeast can improve the antioxidant capacity and immune traits, and liver health of broilers under HSD. Based on the results of the linear regression analysis, the optimal supplementation of Cr-yeast in antioxidant capacity, immunity ability and liver health were at the range of 425.00–665.00, 319.30–961.00, and 800.00–1531.60 µg Cr/kg, respectively.

## 1. Introduction

Intensive feeding is becoming more and more common in the broiler industry. Although high stocking density (HSD) can improve the yield of broilers per unit area to some extent, it reduces the health status of individual broilers [1]. HSD reduced feed intake, body weight gain and carcass weight [2,3]; at the same time, HSD reduced the antioxidant capacity of the serum and tissue of broilers [4,5,6], and then caused oxidative stress to produce a large number of stress proteins (such as the heat shock protein) and activate oxygen-induced rate-limiting enzymes (such as HO-1), then accelerated the tissue oxidative damages [7] and reduced immunity and caused liver damage [8]. Yarahmadi et al. showed that density stress can increase the proinflammatory cytokine Interleukin-1β (IL-1 β) and tumor necrosis factor-α (TNF-α) contents. The release of proinflammatory factors can mediate inflammation [9]. 

Cr(III) has strong antioxidant activity and is considered to be the preferred mineral in poultry diet [10,11,12]. Studies have shown that Cr(III) has pharmacological effects on rodents [13], and yeast can stimulate digestive enzymes to produce better immune response and higher growth performance [14]. It is well known that organic chromium has lower toxicity and higher bioavailability than inorganic chromium [15,16]. Under conventional conditions, adding organic chromium can promote the growth, fast metabolism and antioxidant capacity of broilers [17,18,19]. Under the condition of heat stress, chromium picolinate (Cr-Pic) supplementation can reduce the level of corticosterone and increase the level of serotonin in broilers, indicating that Cr(III) may have the effect of relieving stress in broilers [20]. In addition, at high stocking density, chromium methionine (Cr-Met) supplementation can improve production performance and humoral immunity in layers [21]. Chromium yeast (Cr-yeast) is a yeast preparation enriched with trivalent chromium, which has no toxicity confirmed by animal toxicology tests [22,23]. Therefore, Cr-yeast can ensure the safety of feed. It has the advantages of a short production cycle, simple operation and low production cost [24,25]. According to the meta-analysis of White and Vincent [26], in recent years, the research range of chromium in different forms and sources in broiler diets is 200–2000 µg Cr/kg under non-stress, and the research range under heat stress is 200–4 × 10^5^ µg Cr/kg. Safwat et al. [27] reported that adding 1000 and 1500 µg Cr/kg Cr-yeast to the diet of Arbor Acres broilers can reduce the feed conversion ratio of 0–21 days old and 0–35 days old under normal conditions; Sahin et al. [20] reported that adding 200, 400, 800, and 1200 µg Cr/kg Cr-Pic to the diet of Ross broilers increased feed intake and decreased serum corticosterone content under heat stress. In addition, there have been many studies on the effects of different organic chromium (Cr-Pic, chromium nicotinate (Cr-Nic), chromium propionate (Cr-Pro) and Cr-Met) on broilers, but few studies were reported on the alleviation of high stocking density stress by organic chromium.

Therefore, this experiment aims to explore the effects of different doses of Cr-yeast on the growth performance, serum biochemistry, serum immunity, antioxidant capacity and liver histopathology of broilers under high stocking density stress, in order to obtain the appropriate dosage of chromium yeast and provide a scientific basis for production practice.

## 2. Materials and Methods

### 2.1. Animals, Experimental Design and Management

A total of 684 1-day-old Arbor Acres male broilers were obtained from Daxiang Farming Group Co. (Shanxi, China) and fed with the same starter diets from 0 to 22 days of age. On day 22, broilers were weighted and randomly divided into six treatments with six replicate pens in each group. The broilers in control groups were fed with a control diet and raised at low stocking density (14 broilers/m^2^, 10 birds per cage, LSD) and high stocking density (20 broilers/m^2^, 14 birds per cage, HSD). Individual cages were 100 cm long × 70 cm wide. The broilers in treatment groups were fed with diets supplemented with 200, 400, 800 and 1600 μg Cr/kg Cr-yeast (Alltech Co. Ltd., Tianjin, China) under high stocking density, respectively. The diets of the control group were all manufactured according to the basic diet formula, and the diets of the group added with yeast chromium were all manufactured on the basis of the basic diet by replacing the rice husk powder in the premix with Cr-yeast. The weight of the rice husk powder was reduced according to the Cr-yeast weight, and the total weight was constant. The experimental period was 23~42 days. The corn-soybean meal basal diet was formulated according to the recommendations of the Arbor Acres broiler guide [28]. The compositions and nutrient levels were shown in Table 1.

The broilers were reared in wire-floored cages, in a controlled rearing room environment. The lighting regimen used was 23 h of light and 1 h of darkness. Feed and water were provided ad libitum during the entire experimental period. The rearing room temperature was consistent at 35 °C (the first 3 days), after which the room temperature was gradually reduced by 3 °C each week until the temperature reached 24 °C. This temperature was maintained until the end of the experiment on day 42. The relative humidity of the rearing room was maintained between 50% and 70%. On days 7 and days 21, all broilers were inoculated with an inactivated Newcastle disease vaccine, and on days 14 and 28, broilers were given an inactivated infectious bursal disease vaccine. Drinking water immunization was adopted. Before immunization, the water supply system and watered dispenser was rinsed with clean water, we stopped feeding water to the broilers for 2 h, and then diluted the vaccine in drinking water until the immunized broilers finished drinking the vaccine water, and then the drinking water was supplied normally.

### 2.2. Dietary Nutrients and Chromium Analysis of Diets

The nutritional value of the experimental diets was analyzed according to the methods of the Association of Official Analytical Chemists [29] for total phosphorus (995.11), calcium (927.02) and crude protein (988.05). Methionine, lysine, threonine, tryptophan and cysteine contents were determined using acid hydrolysis with 6 mol L^–1^ HCl (994.12) and measured using an amino acid analyzer (Hitachi L-8800, Tokyo, Japan). Performic acid oxidation was performed prior to acid hydrolysis to determine the methionine concentration. The Cr content in the diets was determined by inductively coupled plasma mass spectrometry (model 7500, Agilent Technologies, Inc., Palo Alto, CA, USA).

### 2.3. Growth Performance

The body weight and feed consumption per replicate in each group of broilers were recorded on days 23 and 42 of the experiment. The feed intake (FI), body weight gain (BWG), and feed conversion ratio (FCR) of broilers were calculated from days 23 to 42.

### 2.4. Sample Collection

On day 42, after fasting for 12 h, six broilers close to the average body weight of each treatment were selected to collect blood and tissue samples. Blood samples were collected into 5 mL anticoagulant-free vacutainer tubes by taking blood from the vein under the wing, and they were stood for 30 min at room temperature, then centrifuged at 3600× *g* for 15 min. The serum samples were collected and stored at −80 °C for further analysis. After blood sampling, the broiler was bled to death through the carotid artery, slaughtered, and then dissected. The heart, liver, spleen, lung, kidney, thymus, breast, leg, abdominal fat and pancreas were weighted for calculating the organ index. The samples of liver, breast and jejunum were collected and stored at −80 °C for the analysis of antioxidant parameters. 

### 2.5. Organ Index

The organ index of heart, liver, spleen, lung, kidney, thymus, breast, leg, abdominal fat and pancreas were calculated by the following formula:organ index = organ weight (g)/body weight (kg)

### 2.6. Serum Biochemistry

The serum contents of albumin (ALB), aspartate aminotransferase (AST), alanine aminotransferase (ALT), alkaline phosphatase (ALP), total bile acid (TBA), glycocholic acid (CG), glucose (GLU), prealbumin (PA), serum ferritin (SF) and transferrin (TRF) were analyzed using an automatic biochemical analyzer (Hitachi 7160, Hitachi High-Tech Corp., Tokyo, Japan). The corticosterone (CORT) concentrations were analyzed with commercial kits (Shanghai Enzyme-linked Biotechnology Co., Ltd., Shanghai, China) according to the manufacturer’s protocol. The kit number was shown in Appendix B.

### 2.7. Serum Immunity

The serum contents of Interleukin-1β (IL-Iβ), Interleukin-10 (IL-I0), thyroxine (T3) and thyrotropine (T4) were analyzed with commercial kits (Shanghai Enzyme-linked Biotechnology Co., Ltd., China) according to the manufacturer’s protocol. The Tumor necrosis factor-α (TNF-α) were analyzed with commercial kits using commercial kits (Nanjing Jiancheng Bioengineering Institute, Nanjing, China) according to the manufacturer’s protocol. The kit number was shown in Appendix B.

### 2.8. Antioxidant Capacity

On day 42, the tissue samples of liver and breast were stored at −80 °C. Then, the tissue samples were used to detect total antioxidant capacity (T-AOC), total superoxide dismutase (T-SOD), glutathione peroxidase (GSH-Px), catalase (CAT) enzyme activities and malondialdehyde (MDA) contents with commercial kits (Nanjing Jiancheng Bioengineering Institute, China) according to the manufacturer’s protocol. The serum contents of T-AOC, SOD, CAT, GSH-Px and MDA were measured by commercial kits (Shanghai Enzyme-linked Biotechnology Co., Ltd., Shanghai, China). The kit number was shown in Appendix B.

### 2.9. Histopathological Analysis of Liver

On day 42, portions of liver were fixed in 10% formalin buffer solution for 48 h, then followed by dehydration, clearing and paraffin embedding procedures. Paraffin sections of 3 μm thickness were stained with hematoxylin-eosin (H&E). Light microscopic examination for histopathological analysis was conducted under an optical microscope (Nikon Eclipse E100, Nikon Co, Tokyo, Japan). Then, the portions were observed under a 200 times microscope, and the severity of the hepatocyte edema and inflammation was scored: for inflammatory cell infiltration, the samples are divided into 0–3 scores (0: none; 1: 2 lesions per 200 times field of vision; 2: 2–4 lesions per 200 times field of vision; 3: 4 lesions per 200 times field of vision); for liver cell swelling, the samples were divided into 0~2 scores (0: none; 1: 1–4 vacuolar cells; 2: ≥ 5 obvious vacuolar cells) [30]; finally, the two scores are added to achieve the total histopathological score.

### 2.10. Statistical Analysis

All data were analyzed by a one-way analysis of variance (ANOVA) for a completely randomized design using the SPSS 26.0. The differences among treatments were determined with Duncan’s Multiple Range test. The data of the low-density control group and high-density control group were analyzed with SPSS 26.0 for an independent sample *t*-test. The linear and quadratic effects of Cr as Cr-yeast levels in high stocking density groups were assessed using orthogonal polynomials. Then, we logged in to SAS to carry out broken-line regression analysis of the principal component indicators to evaluate the optimal addition level of chromium yeast [31]. The pen served as the experimental unit. A probability level of *p *< 0.05 was considered statistically significant.

## 3. Results

### 3.1. Dietary Nutrients and Chromium Analysis of Diets

The analyzed dietary nutrients and Cr contents were presented in Appendix A. The Cr contents in control groups of high stocking density and low stocking density was 468.47 µg/kg. The analyzed Cr concentrations in the diet of HSD + 200, 400, 800 and 1600 µg Cr/kg Cr-yeast groups were 668.47, 834.99, 1204.09 and 2024.63 µg/kg, respectively. The dietary nutrients and the Cr contents in diets supplemented with different levels of Cr-yeast agreed with the expected values.

### 3.2. Growth Performance

The effects of Cr-yeast on the growth performance of broilers on day 23–42 under high stocking density was provided in Table 2. There were no significant differences in BW, BWG and FCR among all groups of broilers on day 23–42. A quadratic significance in FCR was observed in all HSD groups (*p* = 0.037)

### 3.3. Organ Index

The effects of Cr-yeast on the organ index of broilers on day 23–42 under high stocking density was presented in Table 3. There were no significant differences in organ index of heart, spleen, lung, kidney, thymus, breast, leg, abdominal fat and pancreas among all groups (*p *> 0.05). However, compared with the LSD group, the liver index was significantly decreased in the HSD group (*p* = 0.031). Diets supplemented with 200, 400, 1600 µg Cr/kg Cr-yeast significantly increased the liver index of broilers under high stocking density, compared with the HSD group (*p* < 0.05). A linear significance and a quadratic significance in liver index and lung index were observed at different levels of Cr-yeast in the HSD groups (*p* < 0.05).

### 3.4. Serum Biochemistry Parameters

Table 4 showed the effects of Cr-yeast on the serum biochemical parameters of broilers on day 42 under high stocking density. Compared with the LSD group, the contents of serum CORT, TBA, GLU and ALT, ALP activities were significantly elevated, whereas contents of ALB, TP, PA and TRF were significantly decreased in the HSD group (*p* < 0.001). There were no significant differences in AST activities and the contents of CG and SF between the LSD and the HSD group (*p* > 0.05). Compared with the HSD group, diets supplemented with different levels of Cr in the form of Cr-yeast significantly decreased the activities of ALB, ALT and AST (*p* < 0.05). Compared with the HSD group, the contents of CORT were decreased in the HSD + 200 and 400 groups but increased in the HSD + 800 and 1600 groups (*p* < 0.05); ALP activities were elevated in the HSD + 200 and 400 groups but decreased in the HSD + 800 group (*p* < 0.05); TP contents were much higher in the HSD + 200, 800 and 1600 groups (*p* < 0.05); TBA levels were significantly decreased in the HSD + 400, 800 and 1600 groups (*p* < 0.05); CG contents were decreased in the HSD + 200 and 800 groups (*p* < 0.05); GLU contents were decreased in the HSD + 200, 400 and 800 groups (*p* < 0.05); PA contents were much higher and the SF contents were much lower in the HSD + 400, 800 and 1600 groups (*p* < 0.05); and TRF levels were elevated in the HSD +200 and 400 groups (*p* < 0.05). Furthermore, there were no significant differences in the contents of CORT, CG, GLU and PA between the LSD and the HSD + 400 group (*p* > 0.05). Compared with the LSD group, the contents of SF had no significant difference in the HSD + 800 group (*p* > 0.05). The linear and quadratic significances in all serum biochemical parameters were observed at different levels of Cr-yeast in the HSD groups (*p* < 0.05).

### 3.5. Serum Immunity Parameters

Figure 1 showed the effects of Cr-yeast on the serum immunity of broilers under high stocking density. The levels of inflammatory factors in the serum was shown in the Appendix A. There were no significant differences in TNF-α, IL-1β and IL-10 contents between the LSD group and the HSD group (*p* > 0.05). Compared with the LSD group, T3 contents decreased (*p* = 0.014), and T4 contents increased significantly in the HSD group (*p* = 0.011). Compared with the HSD group, TNF-α contents significantly decreased in the HSD + 800 group (*p* < 0.05); IL-1β contents increased in the HSD + 200, 400 and 1600 groups and decreased in the HSD + 800 group (*p* < 0.05); IL-10 contents were much higher in the HSD + 800 group and much lower in the HSD + 200, 400 and 1600 groups (*p* < 0.05); T3 contents significantly increased in the HSD + 200, 400 and 800 groups and decreased in the HSD + 1600 group (*p* < 0.05); T4 contents significantly decreased in the HSD + all Cr-yeast groups (*p* < 0.05). Compared with the LSD group, IL-1β contents decreased significantly (*p* < 0.05), IL-10 and T3 contents increased significantly in the HSD + 800 group (*p* < 0.05), and T4 contents decreased significantly in the HSD + 200 and 400 groups (*p* < 0.05). Serum immunity parameters at different levels of Cr-yeast of the HSD groups showed linear and quadratic significances (*p* < 0.05). 

### 3.6. Antioxidant Status in Serum and Tissues

The effects of Cr-yeast on the antioxidant index of broilers under high stocking density was provided in Table 5. In the serum, compared with the LSD group, T-AOC and GSH-Px activities in the HSD group decreased significantly (*p* < 0.001). There were no significant differences in SOD, CAT activities and MDA contents between the LSD and the HSD group (*p* > 0.05). Compared with the HSD group, the T-AOC significantly increased in the HSD + 200 and 1600 groups (*p* < 0.05), GSH-Px activities were much higher in the HSD + 400 and 1600 groups (*p* < 0.05), SOD activities significantly decreased in the HSD + 200 and 1600 groups (*p* < 0.05), the addition of 200, 400 and 800 Cr-yeast significantly decreased the CAT activities of the HSD group (*p* < 0.05), and the MDA contents significantly decreased in the HSD + 200 and 400 groups (*p* < 0.05). Compared with the LSD group, the T-AOC activities showed no differences in the HSD + 200 and 1600 groups (*p* > 0.05), and the GSH-Px activities had no significant differences in the HSD + 400 group (*p* > 0.05). All antioxidant parameters at different levels of Cr-yeast of the HSD groups showed linear significance and quadratic significances (*p* < 0.05).

In the liver, the MDA contents, the T-SOD and GSH-Px activities showed no significant differences in all indexes between the LSD and HSD group (*p* > 0.05). Compared with the LSD group, the T-AOC and the CAT activities significantly decreased in the HSD group (*p* = 0.031, 0.043). Compared with the HSD group, T-AOC, T-SOD and CAT activities increased in the HSD + 400 group (*p* < 0.05), the MDA contents significantly decreased in the HSD + 200 group (*p* < 0.05), and the GSH-Px activities increased in the HSD + 400 and 1600 groups (*p* < 0.05). Compared with the LSD group, the T-AOC contents increased in the HSD + 400 group (*p* < 0.05). At different levels of Cr-yeast of the HSD groups, T-AOC and CAT activities had quadratic significances (*p* < 0.05), T-SOD and GSH-Px activities showed linear significances (*p* < 0.05), and MDA contents showed linear significance and quadratic significances (*p* < 0.05).

In the breast, the MDA contents and the GSH-Px activities showed no significant differences in all groups (*p* > 0.05), and CAT activities showed no significant differences between the LSD group and the HSD group (*p* > 0.05). Compared with the LSD group, T-AOC contents and T-SOD activities significantly decreased in the HSD group (*p* = 0.026, 0.038). Compared with the HSD group, T-AOC contents significantly increased in the HSD + 400 group (*p* < 0.05), T-SOD activities increased in the HSD + 400, 800 and 1600 groups (*p* < 0.05), CAT activities were no different in all of the HSD + Cr-yeast groups (*p* > 0.05), but CAT activities of the HSD + 800 group were higher than the HSD + 200 group (*p* < 0.05). Compared with the LSD group, T-AOC contents increased significantly in the HSD + 400 group (*p* < 0.05), and T-SOD activities increased in the HSD + 400 group (*p* < 0.05). T-AOC, T-SOD and CAT activities showed linear significances and quadratic significances at different levels of Cr-yeast of the HSD groups (*p* < 0.05).

### 3.7. Liver Histopathological Analysis

Figure 2 and Figure 3 showed the effects of Cr-yeast on the histopathological analysis and score results of the broilers’ liver under high stocking density. Compared with the LSD group, the HSD group had more inflammatory cells and hepatocyte edema, and the total histopathological score significantly increased (*p* = 0.013). Compared with the HSD group, the HSD + 800 group had almost no inflammatory cells and the hepatocyte edema decreased, and the total histopathological score significantly decreased (*p* < 0.05).

### 3.8. Dose Effect Analysis

The optimal addition level of Cr-yeast in broilers under high stocking density was provided in Appendix A. In liver health, the optimal addition level of Cr-yeast was 800.00–1531.60 µg Cr/kg; when the addition level of Cr-yeast was from 319.30 to 961.00 µg Cr/kg, the serum immunity achieved better effects. In antioxidant status, the optimal addition level of Cr-yeast was 425.00–665.00 µg Cr/kg.

## 4. Discussion

Organic chromium can significantly promote animal growth, which has important practical significance in the production and application of livestock and poultry. In the conventional feeding environment, adding 300 and 400 µg Cr/kg Cr-yeast or 400 µg Cr/kg Cr-Pro to the diet could increase the BWG and reduce the FCR of broilers [17,32]. In the heat stress environment, adding 1000 µg Cr/kg Cr-Met or 1500 µg Cr/kg Cr-Nic to the diet can effectively alleviate the heat stress and improve the feed intake and body weight gain of broilers [33,34]. However, in the current study, the addition of different levels of Cr-yeast had no significant effect on the growth performance of broilers under HSD. This is similar to previous studies. Rajalekshmi al [35] found that the addition of 100, 200, 400, 800, 1600, and 3200 µg Cr/kg Cr-Pro had no significant effect on the feed intake of broilers under conventional feeding conditions; the addition of 400 µg Cr/kg organic chromium (Cr-Met, Cr-yeast, Cr-Nic, Cr-Pic) in the diet had no significant effect on ADG, ADFI and FCR [19]. In addition, Ebrahimzadeh et al. [36] reported that adding 200, 400 and 800 µg Cr/kg Cr-Met to the diet had no significant effect on the feed intake of broilers under heat stress; their study pointed out that when 500 and 1000 µg Cr/kg Cr-Met were added to the diet, it significantly increased feed intake and body weight gain of broilers under heat stress [33]. This difference in growth performance results may be caused by differences in stress conditions and chromium levels. 

The immune organ index can be used to evaluate the immune function of broilers. In our study, the HSD significantly reduced the liver index of broilers. Qaid et al. [37] also found that the relative weight of liver in broilers fed with high density was significantly lower than that in broilers fed with low density. During stress, the body produces a large number of glucocorticoids to activate glucocorticoid receptors to play an anti-inflammatory function, so as to start the resistance response of the internal environment to stimulating factors. When stress is excessive, the expression and function of glucocorticoid receptors are often down regulated, resulting in glucocorticoid resistance, leading to systemic inflammatory response and immune organ dysfunction. [38,39,40]. When we added 200, 400 and 1600 µg Cr/kg Cr-yeast to the diet, the significant decrease in broiler liver index in HSD was alleviated. Sahin et al. [41] reported that the addition of 250 µg Cr/kg Cr-Pic under heat stress increased the liver and spleen index of broilers; Jahanian et al. [34] found that the addition of Cr-Met under heat stress increased the thymus and bursa index of broilers. Part of the positive effect of Cr (III) on immune organ index may be due to its anti-inflammatory and antioxidant functions [36].

The stress response is mainly realized through the hypothalamus pituitary adrenocortical axis (HPA), and the increase in CORT content could be used as a marker of stress response [38]. In the current study, the HSD group had much higher CORT contents in serum than the LSD group, which means the high stocking density induced the stress situation of broilers. In addition, Ma et al. [8] reported that the CORT level was significantly higher in the HSD group than the NSD group. In our study, supplementing 400 µg Cr/kg Cr-yeast to the diet of broilers under high stocking density had the lowest contents of serum CORT, which indicated that Cr-yeast could alleviate stress for broilers. The liver is an important metabolic and immune organ. When hepatocytes are damaged, ALT and AST mainly existing in hepatocytes will escape from the cells due to the destruction of the cell membrane and mitochondrial structure, and then enter the blood, so as to increase the activities of ALT and AST in the serum [42]. Most of the ALP in the serum comes from the liver and bone, and mainly exists in the capillary bile duct cell membrane of the liver. When the hepatocytes are damaged, the capillary bile duct cell membrane in the liver is damaged, causing the capillary bile duct cell membrane to move to the basal side, and then ALP leaks into the serum, raising the concentration of ALP in the serum [43]. TBA are synthesized by the liver and secreted into the intestine, part of which is reabsorbed by the intestine, then through the portal vein, and it is finally absorbed back by the liver [44]. When hepatocytes are damaged, bile acids that are reabsorbed from the intestine cannot be effectively absorbed, and the concentration of bile acids in the serum will be increased, so that the level of TBA in the serum can reflect the injury of liver parenchyma [44,45]; PA is a glycoprotein synthesized by the liver and is secreted into the blood. When the liver is damaged, the synthesis and secretion function will be reduced, and the concentration of PA in the blood will be reduced [46]; TRF is mainly synthesized by hepatocytes. The occurrence of liver disease is often accompanied by the decrease in TRF [47]. In this study, HSD increased the activities of ALT, ALP and the contents of TBA, and decreased the levels of PA and TRF in the serum, indicating that the HSD caused damage to the liver. When we added different levels of Cr-yeast to the diet of broilers in the HSD, the activities of ALT and AST in the serum decreased; when Cr-yeast was added at 800 µg Cr/kg level, the ALP activities and CG contents decreased significantly; when Cr-yeast was added at 400, 800 and 1600 μg Cr/kg level, the TBA contents decreased significantly and PA contents increased significantly; when Cr-yeast was added at 200 and 400 µg Cr/kg levels to the diet, the contents of TRF increased significantly. The results showed that the addition of Cr-yeast alleviated the liver damage of broilers under HSD.

TNF-α is secreted by activated T cells; it is an effective pro-inflammatory and immunomodulatory cytokine, which is related to the inflammatory state [48]. As a key pro-inflammatory cytokine, IL-1-β participated in a variety of autoimmune inflammatory reactions and cell activities [49], and IL-10 can stimulate B cell maturation and antibody production to inhibit inflammation [50]. The decrease in T3 represents less thyroid hormone secretion and hypothyroidism, and the occurrence of hypothyroidism is related to the decrease in self-immunity. A large number of studies have proved that heat stress can reduce the contents of T3 and T4 in the serum of broilers [51,52]. Excessive thyroid hormone secretion in this study may lead to the increase in the thyroid hormone [53]. Compared with the LSD, the T3 content of the serum in broilers decreased significantly and T4 increased significantly in the HSD. This phenomenon proved that HSD was the reason for the decrease in broiler immunity. We found that the addition of 800 µg Cr/kg Cr-yeast to the diet significantly reduced the TNF-α and IL-1-β contents of serum in broilers at HSD, and significantly increased the IL-10 level of serum; when we added 400, 800, 1600 µg Cr/kg Cr-yeast to the broiler diet at HSD, the decrease in T3 and the increase in T4 were alleviated, so that the addition of Cr-yeast can inhibit the occurrence of inflammation and enhance the immunity to a certain extent.

The body produces excessive free radicals in the oxidation state, which has strong oxidation and can damage the body’s tissues and cells. SOD and GSH-Px can prevent cell oxidative damage by scavenging free radicals, SOD can catalyze the disproportionation of free radicals, remove excessive superoxide anion (O^2−^) in the body and inhibit the cascade reaction of oxygen free radicals [54]; GSH-Px is an important peroxidase, which can improve the ability of the body to decompose peroxide products [55]. In this study, HSD decreased T-AOC contents and GSH-Px activities in the serum, and T-AOC contents and T-SOD activities in the breast. This is consistent with the research results of Simitzis et al. [56] and Simsek et al. [57]. It showed that the HSD has produced oxidative stress in broilers. In the HSD, the addition of 200 and 1600 μg Cr/kg Cr-yeast in the diet increased the content of T-AOC, and adding 400 μg Cr/kg Cr-yeast increased the content of GSH-Px in serum. Adding 400 μg Cr/kg Cr-yeast increased the activities of T-SOD and the content of T-AOC in the breast. In addition, although HSD did not change the antioxidant index of broiler liver, 400 μg Cr/kg Cr-yeast can increase the T-AOC contents and the activities of T-SOD and GSH-Px in the broiler liver under HSD. Therefore, Cr-yeast may have strong antioxidant activity and could improve the antioxidant capacity of broilers.

Through histopathological analysis, we found that HSD increased inflammatory factors in hepatocytes and serious tissue cavitation. It is known that hepatocytes contain a large number of mitochondria, which are the main place for biological redox reactions. Therefore, the liver not only produces a large amount of ROS, but also is the main organ attacked by ROS. Oxidative stress in livestock and poultry can cause liver structural damage and functional disorder. It is also an important pathological feature of hepatitis, fatty liver, liver fibrosis and other diseases [58]. The results showed that HSD made broilers produce oxidative stress, resulting in liver inflammation, which is consistent with the research results of others [58,59]. When 800 μg Cr/kg Cr-yeast was added to the diet of high-density broilers, there were almost no liver inflammatory factors, which also showed that yeast chromium had a certain anti-inflammatory effect and alleviated liver injury.

In recent years, the research dose range of chromium in different forms and sources in broiler diets is 200–2000 µg Cr/kg under non-stress, and the research range under heat stress is 200–4 × 10^5^ µg Cr/kg [26]. However, there are few studies on the recommended dose. The role of chromium in different body functions may be different. The range obtained in this study is wide, and further research may be needed to narrow the supplementation range.

## 5. Conclusions

High stocking density decreased the antioxidant abilities in the serum and tissues, reduced the immunity and caused liver damage to broilers. Therefore, supplemental 800.00–1531.60 µg Cr/kg in the form of chromium yeast may alleviate the liver damages; supplemental 425.00–665.00 µg Cr/kg in the form of chromium yeast may improve antioxidant parameters; and supplemental 319.30–961.00 µg Cr/kg in the form of chromium yeast may improve the immune status of broilers in high stocking density. This study proves that the addition of Cr-yeast can indeed alleviate the high-density stress of broilers, and provides corresponding suggestions on the appropriate additional range of yeast chromium for effectively alleviating the high-density stress of broilers, and provides a scientific basis for the actual production of broilers under intensive breeding.

## Figures and Tables

**Figure 1 animals-12-02216-f001:**
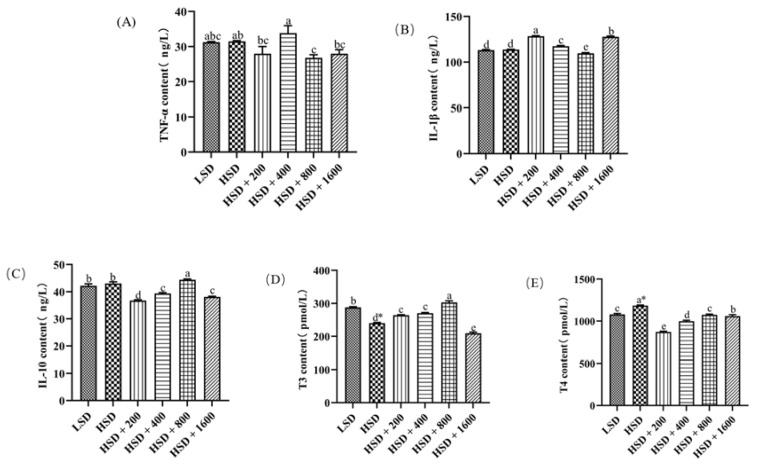
Effects of Cr-yeast on serum immunity index of broilers under high stocking density. (**A**) TNF-α: tumor necrosis factor—α; (**B**) IL-1β: Interleukin-1β; (**C**) IL-10: Interleukin-10; (**D**) T3: thyroxine; (**E**) T4: thyrotropine.; LSD: low stocking density; HSD: high stocking density; Date was shown as the mean ± SE from six independent experiments. * Means *t*-test results of independent samples in group LSD and group HSD (* *p* < 0.05). ^a–e^ These bars without the same letter indicate differences significant at *p *< 0.05.

**Figure 2 animals-12-02216-f002:**
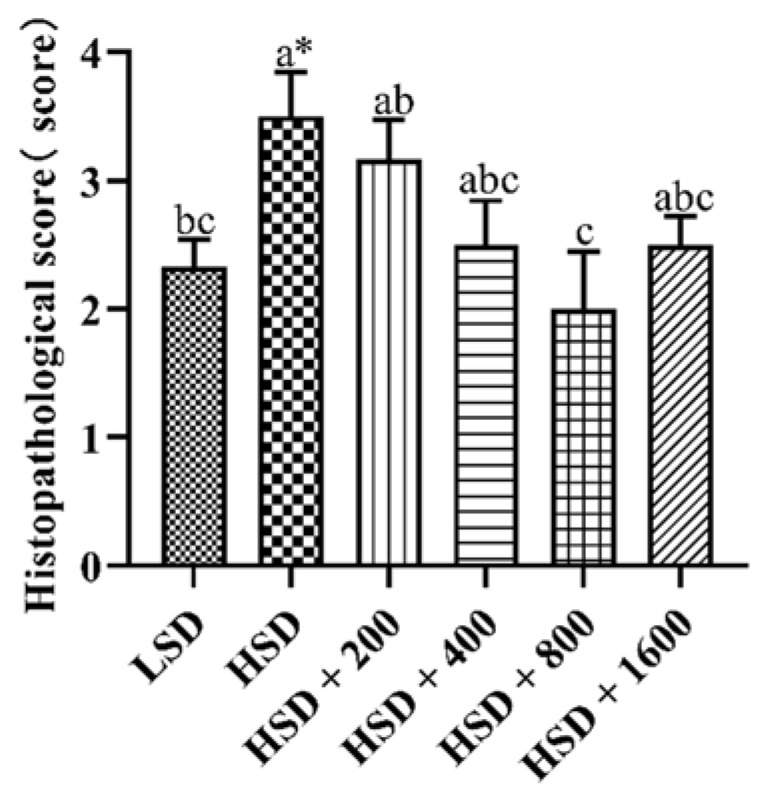
Effects of Cr-yeast on liver histopathological score of broilers under high stocking density. LSD: low stocking density; HSD: high stocking density; Date was shown as the mean ± SE from six independent experiments. * Means *t*-test results of independent samples in group LSD and group HSD (* *p* < 0.05). ^a–c^ These bars without the same letter indicate differences significant at *p *< 0.05.

**Figure 3 animals-12-02216-f003:**
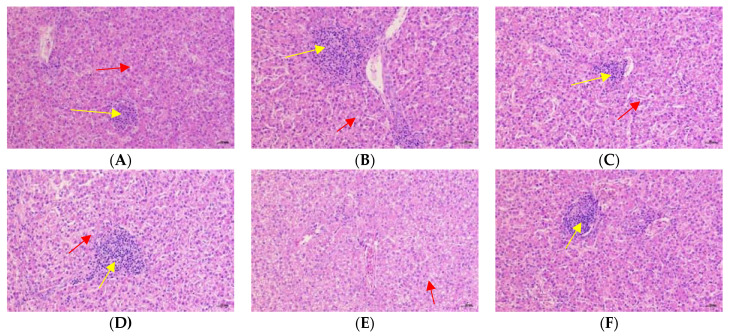
Effects of Cr-yeast on histopathological analysis results of liver of broilers under high stocking density. The magnifications are 200×. The yellow arrows point to inflammatory cells and the red scissors point to the edema of hepatocytes. (**A**) the LSD group; (**B**) the HSD group; (**C**) the HSD + 200 Cr-yeast group; (**D**) the HSD + 400 Cr-yeast group; (**E**) the HSD + 800 Cr-yeast group; (**F**) the HSD + 1600 Cr-yeast group. LSD: low stocking density; HSD: high stocking density.

**Table 1 animals-12-02216-t001:** The ingredients and nutrient levels of the basal diet (%, as fed basis).

Items	Grower Diets (23~42 Days of Age)
Ingredients	
Corn	60.73
Soybean meal	22.70
Corn gluten meal	6.00
Wheat bran	2.00
Soybean oil	4.10
DL- Methionine	0.22
L-Lysine sulphate	0.60
Threonine	0.03
Sodium chloride	0.30
Choline chloride (50%)	0.20
Trace mineral premix ^1^	0.20
Vitamin premix ^2^	0.02
Dicalcium phosphate	1.70
Limestone	1.20
Nutrient levels ^3^	
ME (kcal/kg)	3110
Crude protein	19.05
Lysine	1.12
Methionine	0.53
Threonine	0.71
Tryptophan	0.19
Calcium	0.93
Available phosphorus	0.41
Methionine + cysteine ^3^	0.80

^1^ The trace mineral premix provided per kg of diet: Cu, 16 mg; Zn, 110 mg; Fe, 80 mg; Mn, 120 mg; Se, 0.3 mg; I, 1.5 mg; Co, 0.5 mg. ^2^ The vitamin premix provided per kg of diet: vitamin A, 10,000 IU; vitamin D_3_, 2400 IU; vitamin E, 20 mg; vitamin K_3_, 2 mg; vitamin B_1_, 2 mg; vitamin B_2_, 6.4 mg; vitamin B_6_, 3 mg; vitamin B_12_, 0.02 mg; biotin, 0.1 mg; folic acid, 1 mg; pantothenic acid, 10 mg; nicotinamide, 30 mg. ^3^ Metabolizable energy was calculated values and the others were analyzed values.

**Table 2 animals-12-02216-t002:** Effects of Cr-yeast on growth performance of broilers on day 23–42 under high stocking density ^1^.

Items ^2^	LSD	HSD	HSD + Cr-Yeast (μg Cr/kg)				SEM	*p*-Value ^3^		
			200	400	800	1600		ANOVA	Linear	Quadratic
Day 23 BW (g)	752.02	732.50	735.83	741.25	730.00	726.25	2.865	0.111	0.591	0.749
Day 42 BW (g)	2155.59	2137.55	2078.37	2029.47	2068.80	2050.83	14.754	0.080	0.351	0.548
BWG (g)	1403.56	1405.05	1342.54	1288.22	1338.80	1324.58	13.329	0.064	0.125	0.349
FI (g)	2212.39	2167.44	2083.85	2088.1	2042.62	2111.97	20.839	0.192	0.331	0.669
FCR	1.58	1.54	1.50	1.62	1.53	1.59	0.011	0.148	0.076	0.037

^1^ Values are expressed as a means of six replicates per pen. ^2^ LSD, low stocking density; HSD, high stocking density; BW, body weight; BWG, body weight gain; FI, feed intake; FCR, feed conversion ratio; SEM, standard error of means.^3^ Results of linear and quadratic analysis of five high stocking density groups.

**Table 3 animals-12-02216-t003:** Effects of Cr-yeast on organ index of broilers on day 23–42 under high stocking density (g/kg) ^1^.

Items ^2^	LSD	HSD	HSD + Cr-Yeast (µg Cr/kg)				SEM	*p*-Value ^3^		
			200	400	800	1600		ANOVA	Linear	Quadratic
Heart	4.24	3.82	4.08	4.11	3.89	4.17	0.084	0.690	0.728	0.540
Liver	22.48 ^a^	18.48 ^b^ *	22.86 ^a^	23.87 ^a^	20.80 ^ab^	22.90 ^a^	0.559	0.043	0.042	0.021
Spleen	0.87	0.90	0.78	1.04	0.84	1.07	0.042	0.352	0.337	0.216
Lung	4.88	4.44	4.52	5.4	4.35	4.47	0.112	0.071	0.020	0.027
Kidney	4.62	3.87	5.27	5.06	4.54	4.56	0.143	0.084	0.050	0.184
Thymus	3.27	3.03	2.70	2.17	2.37	2.90	0.163	0.406	0.278	0.831
Breast	179.61	188.51	177.61	161.64	178.77	166.06	2.807	0.068	0.096	0.079
Leg	139.66	142.96	132.97	137.22	135.30	131.28	2.159	0.663	0.755	0.583
Abdominal fat	19.43	19.40	16.74	19.93	14.91	17.15	0.652	0.175	0.234	0.132
Pancreas	1.74	1.62	1.59	1.70	2.00	1.69	0.060	0.449	0.374	0.257

^1^ Values are expressed as means of six replicates per pen. ^2^ LSD, low stocking density; HSD, high stocking density; SEM, standard error of means. ^3^ Results of linear and quadratic analysis of five high stocking density groups. * Means t-test results of independent samples in group LSD and group HSD (* *p* < 0.05). ^a, b^ Means in the same row without the same superscripts differ significantly (*p* < 0.05).

**Table 4 animals-12-02216-t004:** Effects of Cr-yeast on serum biochemical parameters of broilers on day 23–42 under high stocking density ^1^.

Items ^2^	LSD	HSD	HSD + Cr-Yeast (µg Cr/kg)				SEM	*p*-Value ^3^		
			200	400	800	1600		ANOVA	Linear	Quadratic
CORT (μg/L)	133.39 ^e^	161.00 ^c^	144.85 ^d^	129.87 ^e^	173.63 ^b^	192.39 ^a^	3.802	<0.001	<0.001	<0.001
ALB (μg/mL)	224.99 ^a^	217.34 ^b^ **	199.35 ^c^	159.25 ^f^	184.06 ^d^	171.10 ^e^	4.021	<0.001	<0.001	<0.001
ALT (ng/L)	102.41 ^d^	134.01 ^a^ **	102.04 ^d^	128.13 ^b^	107.69 ^c^	93.18 ^e^	2.509	<0.001	<0.001	<0.001
AST (ng/L)	188.64 ^a^	186.86 ^a^	169.90 ^b^	153.84 ^c^	168.92 ^b^	149.14 ^c^	2.951	<0.001	<0.001	<0.001
ALP (ng/L)	141.88 ^e^	169.56 ^c^ **	189.05 ^a^	178.55 ^b^	152.74 ^d^	169.03 ^c^	2.695	<0.001	<0.001	<0.001
TBA (μM)	7.81 ^c^	8.72 ^a^ **	8.73 ^a^	7.03 ^d^	8.55 ^b^	6.85 ^e^	0.132	<0.001	<0.001	<0.001
CG (nM)	259.15 ^b^	269.39 ^ab^	231.16 ^c^	266.04 ^b^	224.02 ^c^	277.60 ^a^	3.636	<0.001	<0.001	<0.001
GLU (μM)	111.49 ^c^	135.02 ^a^ **	91.66 ^d^	109.60 ^c^	117.62 ^b^	135.50 ^a^	2.595	<0.001	<0.001	<0.001
PA (μg/mL)	40.08 ^c^	36.35 ^d^ **	36.49 ^d^	40.56 ^bc^	47.26 ^a^	41.07 ^b^	0.623	<0.001	<0.001	<0.001
SF (ng/mL)	150.71 ^bc^	156.31 ^b^	175.52 ^a^	109.87 ^e^	133.37 ^cd^	120.84 ^de^	4.458	<0.001	<0.001	<0.001

^1^ Values are expressed as means of six replicates per treatment. ^2^ LSD, low stocking density; HSD, high stocking density; CORT, corticosterone; ALB, albumin; ALT, alanine aminotransferase; AST, aspartate aminotransferase; ALP, alkaline phosphatase; TP, total protein; TBA, total bile acid; CG, glycocholic acid; GLU, glucose; PA, prealbumin; SF, serum ferritin; TRF, transferrin; SEM, standard error of means. ^3^ Results of linear and quadratic analysis of five high stocking density groups. Means *t*-test results of independent samples in group LSD and group HSD (** *p *< 0.01). ^a–f^ Means in the same row without the same superscripts differ significantly (*p* < 0.05).

**Table 5 animals-12-02216-t005:** Effects of Cr-yeast on antioxidant status of broilers under high stocking density ^1^.

Items ^2^	LSD	HSD	HSD + Cr-Yeast (µg Cr/kg)				SEM	*p*-Value ^3^		
			200	400	800	1600		ANOVA	Linear	Quadratic
**Serum**										
T- AOC (U/mL)	7.86 ^a^	7.18 ^b^ **	7.95 ^a^	6.94 ^b^	7.02 ^b^	7.85 ^a^	0.073	<0.001	<0.001	<0.001
SOD (pg/mL)	40.83 ^ab^	40.80 ^ab^	39.39 ^c^	39.85 ^bc^	41.44 ^a^	33.68 ^d^	0.459	<0.001	<0.001	<0.001
CAT (ng/L)	69.16 ^a^	69.76 ^a^	65.30 ^b^	54.67 ^d^	60.96 ^c^	68.85 ^a^	0.978	<0.001	<0.001	<0.001
MDA (nM)	14.69 ^ab^	14.69 ^ab^	13.14 ^c^	12.73 ^c^	14.24 ^b^	15.07 ^a^	0.168	<0.001	<0.001	<0.001
GSH-Px (pmol/mL)	14.84 ^a^	13.78^c^ **	13.60 ^c^	14.93 ^a^	12.98 ^d^	14.47 ^b^	0.123	<0.001	<0.001	<0.001
**Liver**										
T-AOC (U/mgprot)	1.25 ^b^	0.78 ^cd^ *	1.18 ^bc^	1.93 ^a^	0.98 ^bcd^	0.72 ^d^	0.081	<0.001	0.446	0.001
T-SOD (U/mgprot)	4.84 ^b^	4.69 ^b^	3.40 ^b^	7.39 ^a^	4.99 ^b^	5.28 ^b^	0.324	0.009	0.005	0.223
CAT (U/mgprot)	3.88 ^ab^	2.77 ^c^ *	3.11 ^bc^	4.75 ^a^	3.54 ^bc^	3.28 ^bc^	0.162	0.003	0.189	0.003
MDA (nmol/mgprot)	1.28 ^ab^	1.43 ^ab^	0.48 ^c^	1.11 ^b^	1.73 ^ab^	1.83 ^a^	0.110	0.001	0.008	0.020
GSH-Px (U/mgprot)	31.13 ^b^	28.25 ^bc^	15.95 ^c^	51.35 ^a^	41.33 ^ab^	45.85 ^a^	2.595	<0.001	<0.001	0.505
**Breast**										
T-AOC (U/mgprot)	0.18 ^b^	0.08 ^c^ *	0.12 ^bc^	0.26 ^a^	0.14 ^bc^	0.08 ^c^	0.014	<0.001	0.001	0.004
T-SOD (U/mgprot)	47.58 ^b^	38.16 ^c^ *	37.87 ^c^	62.40 ^a^	54.91 ^ab^	50.37 ^b^	1.730	<0.001	<0.001	<0.001
CAT (U/mgprot)	1.14 ^a^	1.11 ^a^	0.35^c^	0.48 ^bc^	0.96 ^ab^	0.69 ^abc^	0.081	0.008	0.020	0.029
MDA (nmol/mgprot)	1.11	0.79	0.61	0.87	1.29	0.76	0.087	0.223	0.204	0.126
GSH-Px (U/mgprot)	0.69	0.68	0.63	0.93	0.91	0.50	0.049	0.127	0.055	0.131

^1^ Values are expressed as means of six replicates per treatment. ^2^ LSD, low stocking density; HSD, high stocking density; T-AOC, total antioxidant capacity; T-SOD, total superoxide dismutase; CAT, catalase; MDA, malondialdehyde; GSH-PX, glutathione peroxidase; SEM, standard error of means. ^3^ Results of linear and quadratic analysis of five high stocking density groups. * Means *t*-test results of independent samples in group LSD and group HSD (* *p* < 0.05, ** *p *< 0.01). ^a–d^ Means in the same row without the same superscripts differ significantly (*p* < 0.05).

## Data Availability

The data presented in this study are available on request from the corresponding author.

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
