# Peer review of "Dietary Supplemental Chromium Yeast Improved the Antioxidant Capacity, Immunity and Liver Health in Broilers under High Stocking Density"

_animals, 2022, doi:10.3390/ani12172216_

Round 1

Reviewer 1 Report

The manuscript fits well within the scope of the journal.

The Authors have investigated an interesting topic and the theme has been properly described. The objectives of the study were clearly defined.

The Introduction is written concisely and provides sufficient background. The methods have been properly described, and the design of the experiment and statistical methods applied to allow us to make reliable conclusions.

Results are well presented and thoroughly discussed and data interpretation is appropriate.

The manuscript is well written, presented and discussed, and understandable to a specialist readership.

No significant limitations have been detected, whereas the paper presents novel and useful findings. The results have significant practical implications. According to the results obtained, chickens can benefit greatly from supplementing their diet with Cr-yeast, without experiencing any adverse effects. Due to its ability to activate the enzymes responsible for antioxidant protection in animals, Cr-yeast may be beneficial in terms of antioxidant protection in broiler chickens in both low and high density conditions.

In conclusion, I recommend the acceptance of the manuscript for publication after minor revision.

Suggestions for minor revision are provided within the attached text.

All the best and stay safe

Reviewer 2 Report

The manuscript describes a single experiment in which 22-d-old broilers, housed under high stocking density, were fed 5 graded levels (i.e., 0; 200; 400; 800; and 1,600 µg/kg) of Cr-yeast; an additional treatment consisting of control-fed (0 µg/kg Cr-yeast) broilers housed under low stocking density was used for a total of 6 treatments in the experiment. Growth performance between 22 and 43 d of age was measured as were various organ weights and immune variable in plasma, breast muscle, and liver at Day 43 of age.

GENERAL COMMENTS

The dietary treatments consisted of graded levels of added Cr-yeast. However, there is no chemical analysis of the dietary Cr concentrations from each of the treatments. Without such chemical analysis in all 6 diets, I cannot recommend the manuscript for publication. The reader needs to know the actual concentrations of Cr in *all* diets so as to be ensured the effects were due to the Cr and the correct concentration of the Cr. If analyzed and intended Cr concentrations differ “substantialy,” the data need to be analyzed using the analyzed Cr values, not the formulated (intended) values. Also, to ensure the reader that the effects of the dietary treatments were indeed caused by Cr-yeast and not something else, all 6 dietary treatments must be analyzed (by wet chemistry, not NIRS) for key nutrients, such as CP, Na, Ca, P.

There is no description of how, specifically, the dietary treatments were manufactured. Lines 99-103 give limited information. The reader needs to know how the diets were manufactured (e.g., from a single batch of the control diet, to which the Cr-yeast was added or from 6 individually made diets). Moreover, the diet composition in Table 1 shows only the control diet; the reader needs to know what was replaced to add the Cr-yeast. 

The method for statistical analysis of the data is inappropriate. According to Line 179, Duncan’s Multiple Range Test was used, yet the treatments were ordered and consecutive, which means some sort of regression analysis, not multiple comparisons, is warranted. Preferably, orthogonal contrasts should be used (or else simple regression), with the contrasts’ coefficient adjusted for the unequal spacing of the Cr-concentrations.

It does appear that linear and quadratic regressions were used to evaluate the dietary treatments (e.g., Table 2) and there seems to be some explanation of this procedure in Lines 179-181, but the description is unclear and the reference (number 30) refers to the broken-line regression, not really linear or quadratic regression per se. In any case, data tables (e.g., Table 2) do not contain information about the broken-line regression and the determined Cr requirement. 

The data tables show P-values for linear and quadratic effects. However, it is unclear from the description of the statistical analyses (Lines 176-182) if the low-density (LSD) treatment was included in these regression analyses. It should not be! The effects of the control-fed high-density (HSD) treatment should be compared with the LSD treatment using a separate statistical test (e.g., a contrast). 

Line 313-318 refer to the Cr requirement and “Table S1,” which is not available in the manuscript (which it needs to be to have any value). The ranges of the Cr requirements (Line 315-318) are very large, which leads me to question the results (or the method). The reader definitely need more info on the broken-line data and the results. 

The authors need to re-evaluate the precision of the data in the data tables. I doubt the data can be represented with 2 decimals. 

Some of the language (e.g., Lines 120-123, 179-181) are not written in past tense, but rather some instruction-type language?

SPECIFIC COMMENTS

LINE                COMMENT

17                    …under…

19, 30               The range should be separated using a dash (–), not a tilde (~)

24-25               …fed a common diet from 1 to 23 days of age.

26                    …6 replications…

32+                  The abbreviations need to be defined in the abstract (as well as again in the main text of the manuscript). No one reading just the abstract will know what ALT and ALP means…

35,36                Similarly, the histopathological score and liver index need to be described in the abstract. No one will know what those things are…

76+                  What, exactly, is a “research dose” and a “research range?”

Reviewer 3 Report

The manuscript is well written, but some further corrections are needed to improve the manuscript.

Materials and Methods

Line 129: It would have been interesting to examine the blood of the animals also on day 23 to compare the values.

Line 133: How long was the blood stored from collection to analysis?

Line 143-149: What exact assays were used to measure albumin and prealbumin because only the electrophoresis is suitable for the correct measurement in birds.

Line 157: How long were the tissue samples stored?

Are all laboratory tests validated for broiler chickens?

Line 176: What tests were used to verify normal distribution of the analytes?

Results

Line 200: Please provide the exact calculated p-value here and throughout the manuscript.

Line 215: Only enzymes show activity

Table 4: Please use SI units.

Discussion

Some parts of the discussion are more part of the introduction and could be shortened or removed.

Line 344: Even on the contrary, the animals actually gained less weight. Could it be that the addition of Cr affected the taste of the feed and therefore the animals have fed less?

Line 381: A part of the AST also comes form the muscles, which has to be taken into account especially after capture for blood sampling.

Reviewer 4 Report

A good study looks at using supplementation to improve the health of broilers with high stocking density, which significantly reduces the living conditions, promotes inflammation, and negatively affects growth. 

Line 26: "with six repetitions in each treatment" is a little confusing

Were power calculations done to pic the sample size?

What was the level of Chromium in the control diet?

Where are the levels of anti-inflammatory markers measured in the serum and inflammatory markers such as TNF-alpha?

What would be the future implications or future directions for this study? It would be a good idea to shed some light on the conclusion section of the manuscript 

Round 2

Reviewer 2 Report

The revised manuscript does not adequately address some of my major concerns, explained in my comments to the first (original) submission of the manuscript. I encourage the editors of Animals to work with the authors to correct the issues:

While there appears to be a chemical analysis of the dietary nutrient and Cr concentrations (Lines 134-141), the analyzed values are not shown in the manuscript. Because the treatments were dietary in nature and because the treatment was dietary Cr, these values are essential to show in the manuscript itself, not some supplementary file that readers may or may not have access to. Without analyzed values in a table in the manuscript itself, I cannot recommend the manuscript be accepted for publication. Ditto for Tables S2 and S3—if these values are important enough to measure and describe, why not shown them in the manuscript? 

The revised description of the basal diet (Lines 106-110 and Table 1) is inadequate. Apparently, rice husk powder in the basal diet was replaced with Cr-yeast (Lines 109-110), but no rice husk powder is shown in Table 1! As a result, I have little faith that the experimental diets were indeed manufactured correctly. Sorry, but the *dietary* treatments are essential in a nutrition-research experiment; if you cannot correctly describe the dietary treatments, what else was done incorrectly?

Although the description of the statistical analysis has improved, I am still questioning the methods. Orthogonal contrasts were used (Line 199-200), but no information is given if the contrasts were adjusted for the unequal spacing of the dietary Cr levels (they need to be). Similarly, no information is given about the exclusion of the LSD treatments in regression analysis.  

The authors (or Animals editors) really need to work on the use of English language. I still don’t know what a ‘research dose’ is… I don’t think there is such a word. (eg., Line 497). Some of the language is still not written in past tense, and still in some instruction-type language (e.g., Line 200). The manuscript contains typographical errors. 

My recommendation is to reject the paper based on inadequate revision. I don’t really want to review the manuscript a third time, wasting more of my time.